# Robots for Forest Maintenance

**Tiago Gameiro** [1,*], **Tiago Pereira** [1], **Carlos Viegas** [2], **Francesco Di Giorgio** [3] and **NM Fonseca Ferreira** [1,4]

1    Polytechnic Institute of Coimbra, Coimbra Institute of Engineering, Rua Pedro Nunes-Quinta da Nora, 3030-199 Coimbra, Portugal; tmsp1998pereira@gmail.com (T.P.); nunomig@isec.pt (N.F.F.)
2    Univ Coimbra, ADAI, Department of Mechanical Engineering, Rua Luís Reis Santos, Pólo II, 3030-788 Coimbra, Portugal; carlos.viegas@uc.pt
3    Polytechnic of Torino, Corso Duca Degli Abruzzi, 24, 10129 Torino, Italy; francescodigiorgio26@gmail.com
4    GECAD—Knowledge Research Group on Intelligent Engineering and Computing for Advanced Innovation and Development of the Engineering Institute of Porto (ISEP), Polytechnic Institute of Porto (IPP), 4200-465 Porto, Portugal
*    Correspondence: a2019124292@isec.pt or tiagocostagameiro@gmail.com

**Abstract:** Forest fires are becoming increasingly common, and they are devastating, fueled by the effects of global warming, such as a dryer climate, dryer vegetation, and higher temperatures. Vegetation management through selective removal is a preventive measure which creates discontinuities that will facilitate fire containment and reduce its intensity and rate of spread. However, such a method requires vast amounts of biomass fuels to be removed, over large areas, which can only be achieved through mechanized means, such as through using forestry mulching machines. This dangerous job is also highly dependent on skilled workers, making it an ideal case for novel autonomous robotic systems. This article presents the development of a universal perception, control, and navigation system for forestry machines. The selection of hardware (sensors and controllers) and data-integration and -navigation algorithms are central components of this integrated system development. Sensor fusion methods, operating using ROS, allow the distributed interconnection of all sensors and actuators. The results highlight the system's robustness when applied to the mulching machine, ensuring navigational and operational accuracy in forestry operations. This novel technological solution enhances the efficiency of forest maintenance while reducing the risk exposure to forestry workers.

**Keywords:** mobile robot; ROS; forest maintenance; sensor fusion

## 1. Introduction

Forest-vegetation management is a fundamental direction to take in an effort to reduce the risks associated with forest fires. Recent reports from the European Environment Agency highlights the growing importance of this problem, emphasizing a significant increase in the risk of forest fires across Europe, which is unquestionably related to the effects of climate change [1]. Such effects are linked to the increasing frequency and severity of forest fires in recent years [2]. The rise in heatwaves, primarily caused by the ongoing warming of our planet, resulted in drier environments, fostering the conditions for the occurrence of forest fires. This vicious cycle increases carbon emissions to the atmosphere, intensifying climate change effects, and leading to a cyclic increase in the frequency of forest fires.

Vegetation management through selective removal is a preventive measure which creates discontinuities that will facilitate fire containment and reduce its intensity and rate of spread [3]. However, such a method requires vast amounts of biomass fuels to be removed, over large areas, which can only be achieved using mechanized means, such as forestry mulching machines [4]. Ensuring adherence to safety guidelines is vital for workers in proximity to machines. In addition to standard safety protocols [5], utilizing uncut timber parcels as a physical barrier enhances ground-level worker safety.

The integration of autonomous systems has transformed various sectors of our society in our ever-evolving technological landscape. Their use marked the beginning of a new era of mobile robotics, providing unparalleled safety, precision, and efficiency in various situations [6]. These systems have become indispensable resources in diverse industries, including health care, logistics, agriculture, and the military [7]. Their ability to maneuver and perform tasks that may be dangerous, or complex has been crucial in protecting human involvement in hazardous environments [8]. Furthermore, they are essential for a variety of precision tasks, like continuous operational monitoring in industrial environments, thanks to their exceptional accuracy and continuous operational capability [9].

The use of autonomous systems in the forestry sector has been impaired by multiple challenges including performing operations in unstructured environments [10], implementing navigation systems in forest environments with a lack of visual or structural references [11], and GNSS limitations due to dense vegetation [12], amongst others.

Recently, a few efforts have been made in these areas. Within the SEMFIRE project, a Bobcat was fitted with multiple sensors and adapted to navigate autonomously in an unstructured environment [8].

In a separate application, a real-time LiDAR Odometry and Mapping algorithm was implemented in a forwarder unit (Komatsu Forest 931.1) for autonomous tree mapping in a dense-canopy forest. The methodology entailed creating a 2D topological graph from a point cloud map, identifying trunks through clustering, and establishing correspondences between local and global maps [13].

The AgRob V18 forestry robot, designed specifically for biomass collection, experienced problems with truck movement and caused vibrations that affected IMU data. The researcher extensively studied localization and navigation methods and finally adopted LeGO-LOAM among the SLAM techniques to reach optimal performance. Other robotic applications introduced in this study include a visual navigation UAV for estimating tree diameter using semantic LOAM and a method for estimating mangrove forest biomass using UAV LiDAR data [14].

An ad hoc solution was created using a distributed-sensor approach, placing several optical and laser sensors in various locations throughout the robot structure. We used a small-footprint mobile LiDAR system on an FGI ROAMER R2 vehicle to comprehensively evaluate SLAM-aided stem mapping for the forest inventory. Using the improved maximum likelihood estimation algorithm, the study compared three navigation approaches: GNSS-only, GNSS + IMU, and SLAM + IMU. SLAM was found to be less practical in open areas, but SLAM + IMU showed a 38% higher precision than GNSS + IMU in dense forest regions [15].

These diverse technological applications demonstrate the evolving landscape of robotics in forestry, employing advanced algorithms, artificial intelligence, and sensor technologies for tasks such as inventory operations, biomass estimation, and autonomous navigation in challenging environments. The aim of this work is to introduce a novel modular perception, control, and navigation system for generic forestry machines, comprising a variety of sensors, computers, and dedicated software and algorithms. Forestry robotics encompasses a variety of applications, but one of the most crucial applications involves the removal of excess biomass using forestry shredders. This removal of flammable material in certain locations creates a fuel discontinuity which plays a fundamental role in reducing the frequency and intensity of forest fires. This specialized equipment has been meticulously designed to oversee forestry operations such as the maintenance and management of forest fuel breaks or the defensive space around critical infrastructures [16].

This project, derived from the Forest for Future (F4F) initiative, endeavors to create and showcase autonomous forestry machinery designed for vegetation cutting. Adapting existing remote-controlled forestry platforms from the market involves incorporating both hardware and control software to ensure a secure and efficient autonomous operation, aiming for a Technology Readiness Level (TRL) of at least 7. At the project's conclusion, a user manual will be produced. Implementation requires the installation of various

sensors—vision, infrared, LiDAR, sonar, GPS, accelerometers, and gyroscopes—coupled with electronic components and structural adjustments to the machines. Collaborators in this venture encompass the Engineering Institute of Coimbra of the Polytechnic of Coimbra, Pinhal Maior, Association for the Development of the Southern Interior Pine Forest, and the University of Coimbra [17].

## 2. Materials and Methods

### 2.1. Description of the Forestry Machine

The robotic platform used in this work is a remotely operated and unmanned robot manufactured by MDB (Italy) [18], model LV600 Pro, depicted in Figure 1. This platform possesses a diesel engine to generate the hydraulic pressure required to power the attached tool and both the locomotion hydrostatic motors. An onboard CPU controls the electric valves responsible for operating the several functions of the robot. This is a common configuration for many forestry machines. One of the standout features of this machine is its ability to handle inclinations of up to 60° in all directions. It possesses a robust chassis, and a variety of modular tools can be coupled to it. In this work, a forestry shredder was used, motorized with a hydrostatic motor.

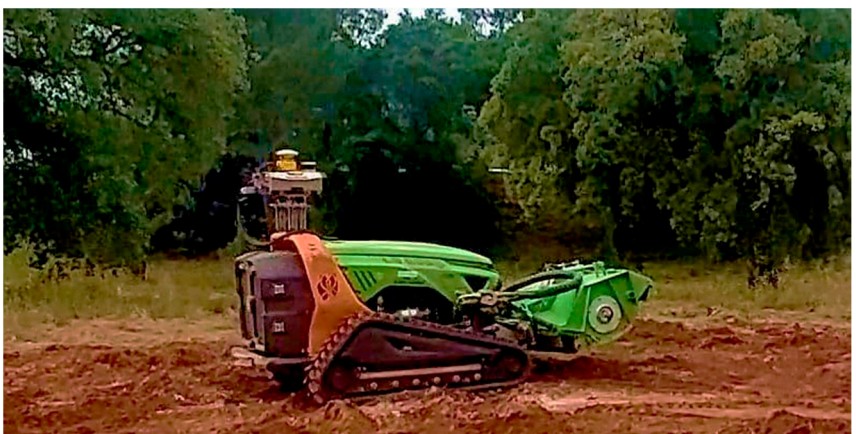

**Figure 1.** Forest Machine MDB-LV600.

### 2.2. Architecture of the Sensor System

The autonomous operation of the robot requires global perception and localization, which can be achieved using multiple sensor systems. The inclusion of various sensors allows the robot to leverage the different capabilities of each, combining their information to gain a more comprehensive, accurate, and reliable view of the surrounding environment [19]. For instance, a robot may use an LIDAR (Velodyne VLP-16), an ultrasound sensor, and an RGB-D camera (Realsens d435i) to perceive the distance to a certain object, depending on their suitability for each situation, for example, where there are low-visibility conditions (potentially affecting vision-based solutions), smoke or debris (affecting laser-based systems), or complex geometry targets (difficult to obtain using ultrasound sensors).

To determine its position, the robot may employ a global navigation satellite system (GNSS) sensor. However, each sensor has specific characteristics, such as its quality, noise level, and sensitivity to environmental conditions. Therefore, a system integrating multiple sensors with sensor fusion must be capable of handling these uncertainties and providing reliable and robust information [20]. In Figure 2, the developed system, named Sentry, is depicted.

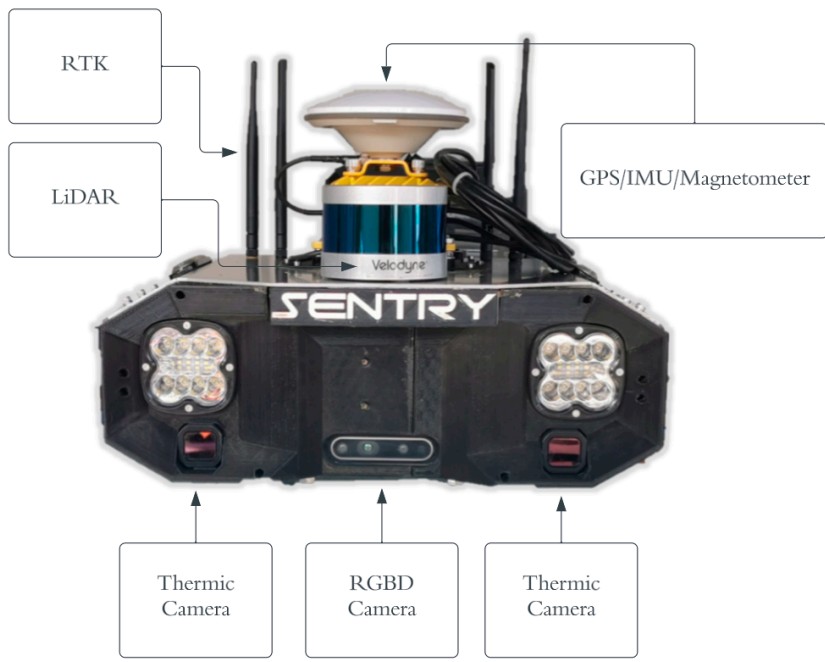

**Figure 2.** Sensorial and Control System (Sentry).

Regarding the hardware in this module, Table 1 specifies each one that as being used inside of Sentry.

**Table 1.** Hardware Components of Sentry.

| Quatity | Hardware | Designation |
|---|---|---|
| 1 | Controller | Arduino Portenta |
| 1 | Computer | NVidea Jetson Xavier NX |
| 1 | Router | RUT360 D-Link |
| 1 | GNSS Receiver/IMU/Magnetometer | Duro Inertial/Bosch BMI160/Bosch BMM150 |
| 1 | RTK | Piksi Multi Evalutation Kit |
| 1 | LiDAR | Velodyne VLP16 |
| 2 | RGBD Camera | Intel Realsense D435I |
| 2 | Thermal Camera | FLIR ADK |
| 1 | Battery | LIFEPO4/12.8 V/48 Ah |

The computer (NVIDIA Xavier NX) is running the Ubuntu 20.04 operating system. All the sensors present communicate through the ROS (Robot Operating System) framework, where the sensors act as slaves and the computer as the master. Table 2 shows the specifications of the computer.

**Table 2.** Jetson Xavier NX Module Specification [21].

| | |
|---|---|
| RAM | 16 GB |
| AI Performance | 21 TOPS |
| GPU | 384 core NVIDIA Volta/48 Tensor Cores |
| CPU | 6-core NVIDIA Carmel ARM®v8.2 64-bit/6MB L2 + 4MB L3 |
| Memory | 128-bit LPDDR4x 59.7GB/s |
| Storage | 1 TB |
| Power | 20 Watts |
| CSI | Up to 6 cameras (36 via virtual channels)/D-PHY 1.2 (up to 30 Gbps) |
| Dimensions | 69.6 mm × 45 mm/260-pin SO-DIMM connector |
| Networking | 10/100/1000 BASE-T Ethernet |

Based on the data acquired by the sensors, the system processes the information comprehensively and formulates its assessment. This process is known as information fusion from multiple sensors, where, from a mathematical perspective, the sensor measurement values constitute a measurement space, and information fusion involves a projection within this space, following specific principles. Information fusion from multiple sensors requires integrating and analyzing information from various sources to make approximations and decisions based on specific principles [22].

The system is designed to perform a broad range of tasks, which can be categorized into two main types: fundamental tasks and functional tasks. Understanding the machine's behavior during operation, such as checking its position, and inclination, and identifying obstacles, is a fundamental task. Functional tasks are related to the actions the system can perform, such as moving the tracks, activating the crusher, and the tool's elevation or descent, among others. To accomplish these tasks, the system is equipped with data acquisition and processing systems. The architecture of the robotic system, as well as the functioning of the entire sensor module, can be observed in Figure 3, where the entire process is summarized using a graph to illustrate the general concept of the system. The '/' behind the word, blue squares, means the topics obtained by the nodes, yellow squares.

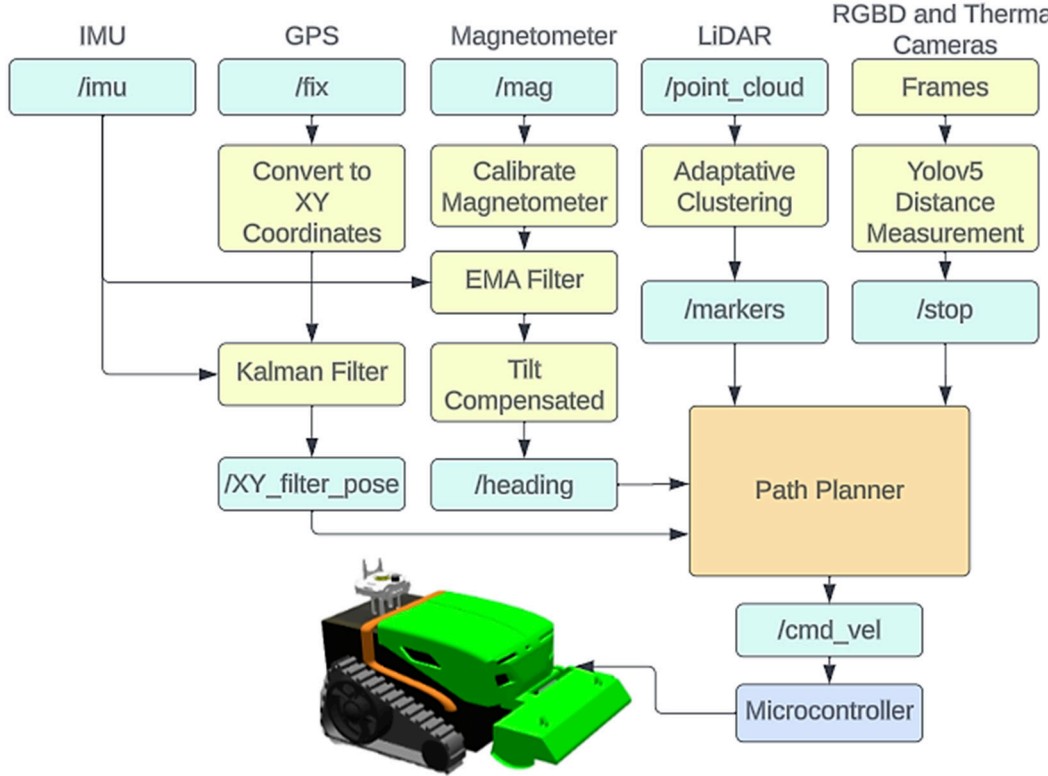

**Figure 3.** Control and Sensor System Architecture.

In Figure 3, the Inertial Measurement Unit (IMU) sensor, which has 6 Degrees of Freedom (6DOF), an accelerometer, and a gyroscope, is used to obtain information about the tractor's behavior. This type of sensor provides information that has been directly acquired, meaning that these data have not been modified, formatted, or processed [23]. Before being used for analyses or other purposes, the data needs to be organized, cleaned, and processed. This type of processing may include noise elimination, error corrections, data format conversions, and transformations of data into structures that are more suitable for future analyses [24]. IMU data were input into a Kalman filter along with GPS coordinates (lat, lon) which were converted to (x, y) to obtain a position in (x, y) with minimal error variation. These IMU data are also used with magnetometer data, after calibrating, in an Exponential Moving Average (EMA) filter to obtain an orientation with tilt compensation.

The Light Detection and Ranging (LiDAR) 3D algorithm provides three-dimensional information using a point cloud, and clustering is performed based on this information. Whenever a cluster of points is detected, it is considered an obstacle, and a box whose dimensions encompass that cluster of points is defined. The positions of these boxes are recorded in a topic designated as /markers.

Both RGBD and thermal cameras use the same YoloV5 algorithm. Thus, besides being able to detect obstacles, they also make it possible to detect the presence of people. However, distance calculation has only been implemented in RGBD cameras. Therefore, whenever a person is detected within 7 m, a publication is made to a topic named "/stop." This topic is subscribed to by the trajectory planning system as an emergency input to stop the robotic system.

### 2.3. System Operation

The robotic system communicates with an online platform controlled by the user, responsible for the tractor. From this interface, the user can access all information about the tractor, from its position to the visualization of all cameras. In other words, all the information available on the robot can be presented in this interface.

The user can outline an area on the map of the online platform, and from this map, the system converts the information into navigable routes. The points on the map are transformed into coordinates and sent to the robot, creating a route to the cleaning area. Additionally, the user can mark the presence of obstacles on the map, serving as restrictions by informing the areas to avoid.

Due to the maneuverability limitations of forestry vehicles, it is essential to reserve an area known as a "maneuvering area" to allow, for example, the turns of the vehicle [25]. The simplest approach is to assign a zone of constant width around the terrain; however, this would allocate a large amount of space for a very reduced utility area. To optimize this situation, it is possible to build maneuvering areas only along the edges of the terrain, where maneuvers effectively occur, thus reducing the area required for this purpose [26]. The working lanes are subsequently created in the inner part of the terrain, which is the region that remains after subtracting the maneuvering areas. In two-dimensional terrains, a reference line can be used as a guide for creating lanes, where each parallel line defines a lane [13].

Figure 4 is a conceptual representation illustrating the creation of planned trajectories, where the user-defined area is presented. If the user does not specify restricted areas, the algorithm will automatically generate a trajectory for robot navigation, as shown in Figure 4b. In the case of defined prohibited areas, as illustrated in Figure 4c, the algorithm creates a path to avoid these areas.

The field partitioning process involves four steps: dividing the field into headland and main area, generating smooth lines for the headland, creating lines for the main area, and calculating viable curves [27]. If the headland width is unspecified, it is calculated based on the working width and machine turning radius. The outcome is a graph structure used for route generation or exporting lines to automatic guidance systems [28].

The online platform, still in development, utilizes the RabbitMQ protocol for communication. RabbitMQ, a free and open-source solution, acts as a message broker using the AMQP. It serves as an intermediary between microservices, ensuring fault tolerance and scalability. This facilitates bidirectional communication between ROS and the web. In development tests, .txt files with Cartesian coordinates were used to represent "goals" for the tractor, providing a means of communication with the PC.

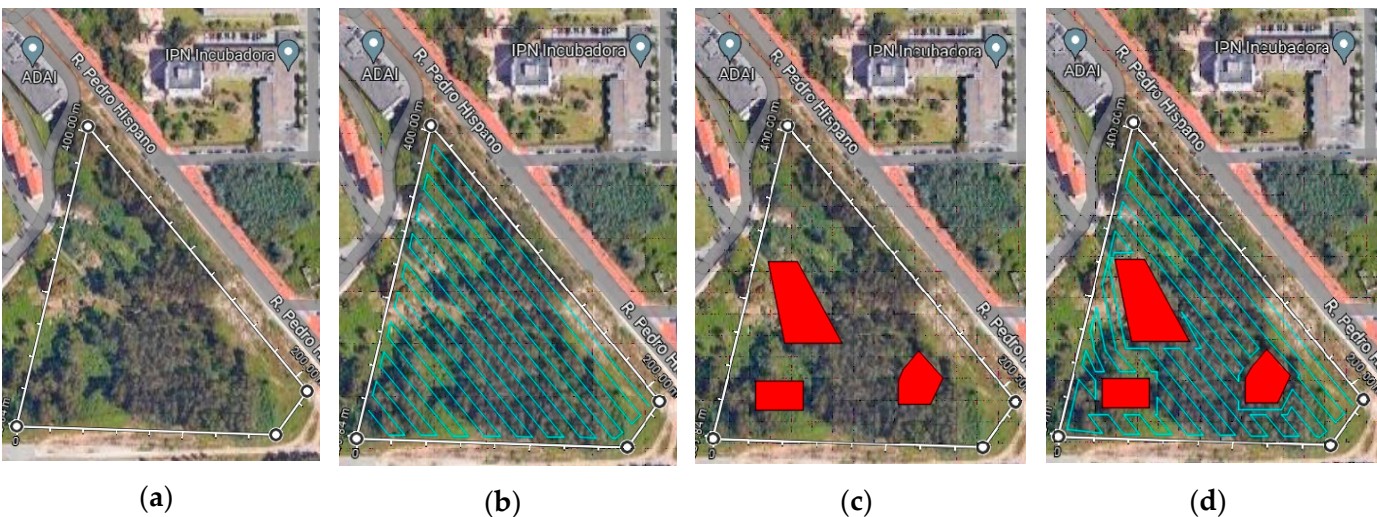

(**a**)  (**b**)  (**c**)  (**d**)

**Figure 4.** Representation of the choice of the area. (**a**) Area selection. (**b**) Path creation. (**c**) Restricted areas. (**d**) New path.

During development, specific rules were identified by observing a professional using this tractor daily for forest thinning to ensure that vegetation is adequately cleared. Figure 5 represents the state diagram describing the machine's operation during the thinning process.

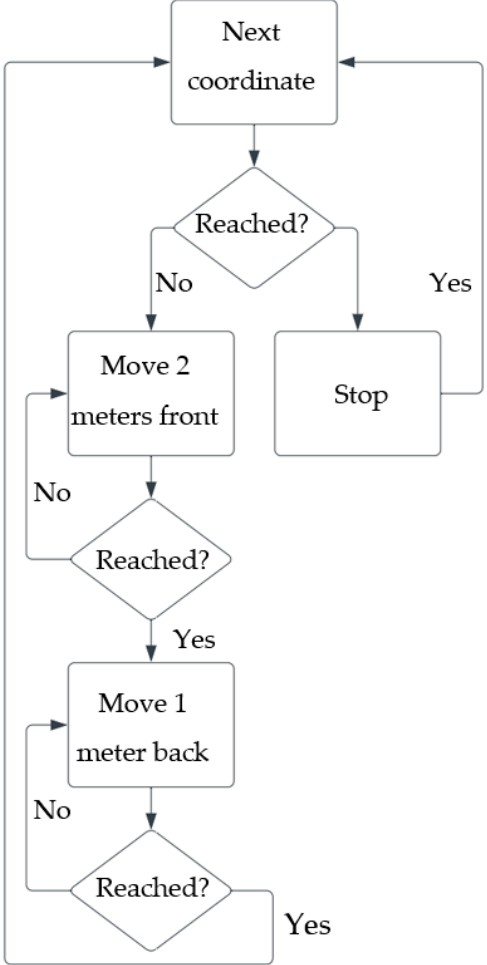

**Figure 5.** State machine for vegetation mulching.

The practice of moving two meters forward and one meter backward plays a crucial role in achieving thorough cleaning. Essentially, the tractor advances two meters forward while clearing vegetation. Subsequently, it retreats one meter, smoothing the terrain with the tool and preventing strain on the equipment. If the tractor were to continually move forward, it would lead to a buildup of biomass, making it challenging for the machine to handle the excessive vegetation. Therefore, the alternating forward and backward movement allows for thinning and leveling intervals, ensuring that the machine operates efficiently without undue stress while leaving the terrain thoroughly cleaned.

### 2.4. Kinematics and Position Representation

While a robot operates in each location, its actions are intrinsically linked to its precise location in that environment, including both its position and orientation. In a simple sense, a robot can be considered a rigid body, where its constituent parts maintain a constant relative position to each other and to the rigid body itself. Referring to Figure 6, we establish a coordinate system in a reference frame (W), serving as the starting point for the robot's navigation. Let us assume that the robot is located at the reference point P, with a new coordinate system (R). In this way, the vector r represents the displacement between the coordinate systems (W) and (R) located at point P. This coordinate system can be defined in either polar or Cartesian coordinates.

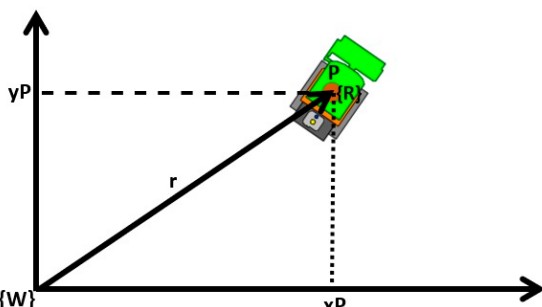

**Figure 6.** Definition of the robot's position in a reference system.

The robot's position concerning the coordinate system (W) is described by the combination of the position (x, y) and the orientation angle θ for the coordinate axes. The forest machine used is a differential-drive robot, and its configuration in a two-dimensional coordinate system is defined by the robot's position along the (x, y) axes and the orientation angle θ concerning the coordinate axes as shown in Figure 7.

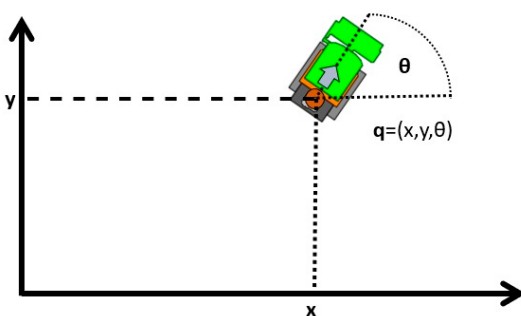

**Figure 7.** Representation of the position of a differential robot.

The vector "q" represents the three distinct components of the robot's position. In our case, neither the tracks nor the combustion engines are equipped with sensors, and despite the knowledge of the robot's kinematics, for navigation purposes, using the robot's position becomes more accurate than the rotation speed of each track. Therefore, to control the speed of each track, different voltage levels are applied depending on the robot's position.

The simulation model is formulated based on the ICR (Instantaneous Center of Rotation) kinematic model, which can simulate the vehicle's movement to its rotation centers. Ideally, these centers should be positioned at the center of each track, but they may shift while navigating diverse terrains [29]. Figure 8 illustrates a schematic representation of the robot's kinematics.

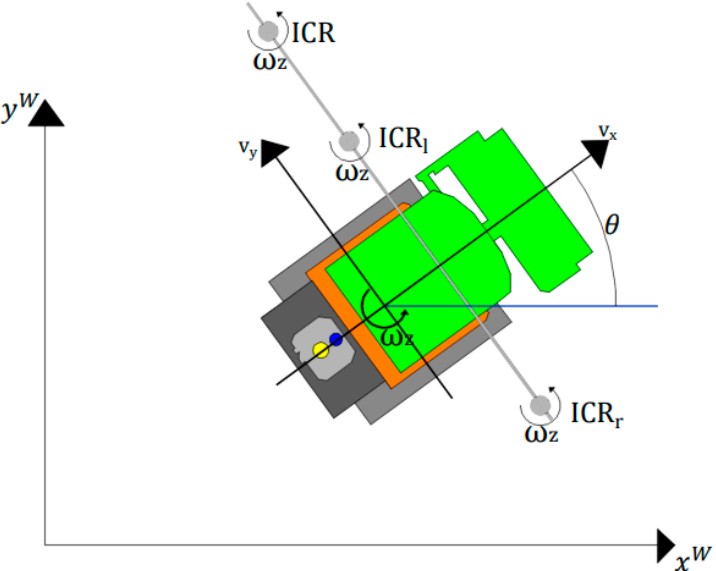

**Figure 8.** Tractor Kinematics.

In the movement of tracked vehicles, the ICR is a pivotal point in the horizontal plane where the vehicle appears to rotate without lateral motion [30]. Unlike when considering the vehicle, it is essential to account for the individual movements of the tracks in contact with the ground. Each track behaves as an additional rigid body with a distinct velocity, leading to different ICRs for the left and right tracks compared to the entire vehicle [31]. This distinction is crucial for understanding the complex motion dynamics of tracked vehicles.

$$ICR_l = (x_{ICRl}, \, y_{ICRl}) \; e \; ICR_r = (x_{ICRr}, \, y_{ICRr}) \tag{1}$$

It is important to emphasize that this definition is associated with the point on the ground where the tracks contact the surface, rather than the rolling axis of the tracks themselves. The local reference coordinates for the vehicle's ICR and the ICRs of the tracks can be geometrically determined through the following functions [30]:

$$x_{ICR} = \frac{-v_y}{\omega_z} \, x_{ICRl} = \frac{V_l - v_y}{\omega_z} \, x_{ICRr} = \frac{V_r - v_y}{\omega_z} \tag{2}$$

$$y_{ICR} = y_{ICRl} = y_{ICRr} = \frac{v_x}{\omega_z} \tag{3}$$

In calculating the inverse functions, it is possible to obtain the instantaneous translational and rotational velocities with respect to the reference system. These velocities represent the direct kinematics of the robot if the ICRs of the tracks are obtained using:

$$v_x = \frac{V_r - V_l}{x_{ICRr} - x_{ICRl}} * y_{ICR} \tag{4}$$

$$v_y = \frac{V_r + V_l}{2} - \frac{V_r - V_l}{x_{ICRr} - x_{ICRl}} * \frac{x_{ICRr} + x_{ICRl}}{2} \tag{5}$$

$$\omega_z = \frac{V_r - V_l}{x_{ICRr} - x_{ICRl}} \tag{6}$$

*2.5. Control*

The robot's control system is of the MIMO (Multiple Inputs and Multiple Outputs) type. This tractor is particularly useful for meeting the specific needs of forestry operations due to the unpredictable and variable nature of these environments. When working in forests, the terrain is often uneven and rugged, with natural obstacles that can hinder movement and maneuvering [8]. The control system consists of two main components: one component related to control types and another control component focusing on the "Microcontroller". Due to the lack of knowledge about the speeds of each track, control is achieved by obtaining the position and orientation of the tractor. By utilizing information from various sensors in the system, both the position and orientation of the robot can provide linear and angular velocities. The input variables in this system are designated as references R1 and R2, as seen in Figure 8. These references are obtained using the tractor's trajectory planning system and serve as input references for each of the controllers that manage the movement of each track. The control of the tractor tracks' electromechanical valves is outlined as per Algorithm 1.

---

**Algorithm 1—Control Algorithm of the Microcontroller**

---

IF R1 = 0 AND R2 = 0
    $VR = VL = 0$
IF R1 = 0 AND R2! = 0
    $VR = VL = abs(MaxRot-MinRot\ 0.9 * R1 + MaxRot - MaxRot-MinRot\ 0.9)$
    IF R1 <= −0.01
    TURN CLOCKWISE ($VR = -VR\ AND\ VL = VL$)
IF R1 >= 0.01
    TURN ANTI-CLOCKWISE ($VR = VR\ AND\ VL = -VL$)
    IF $x$ ! = 0 AND $z$ = 0
    $VR = VL = abs(MaxLinear-MinLinear\ 0.9 * R2 + MaxLinear - MaxLinear-MinLinear\ 0.9)$
    IF R2 <= −0.01
    GO BACK ($VR = -VR\ AND\ VL = -VL$)
IF R2 >= 0.01
    GO FRONT ($VR = VR\ AND\ VL = VL$)
    ELSE
    IF R2 > 0 AND R1 > 0 AND R1 $\leq$ 0.1
    R2 = 1.1 * ($MaxLinear - MinLinear$) * R2 + $MinLinear$
    VR = R2
    $VL = R2 * (1 + 1.2 * z)$
IF $x$ > 0 AND $z$ > 0.1 AND $z \leq$ 0.03
    R2 = 1.1 * ($MaxLinear - MinLinear$) * R2 + $MinLinear$
    $VR = R2 − 4*R1\ 0.3 − 60*R1$
    $VL = R2 − 7*R1\ 0.3 + 60*R1$
IF R2 > 0 AND R1 > 0.3
    $VR = 1.1 * (110 + 10\ 0.7*abs(z-0.3))$
    $VL = 1.1 * (70 + 15\ 0.7*abs(z-0.3))$
    IF R1 > 1
    $VR = MaxLinear$
    $VL = MinLinear$

---

The error variables *e*1 and *e*2 are obtained using the difference between the desired values and the measured values, as we present in Figure 9. Both controllers directly actuate the tractor's propulsion system, thus correcting the movement based on the detected error in each controller.

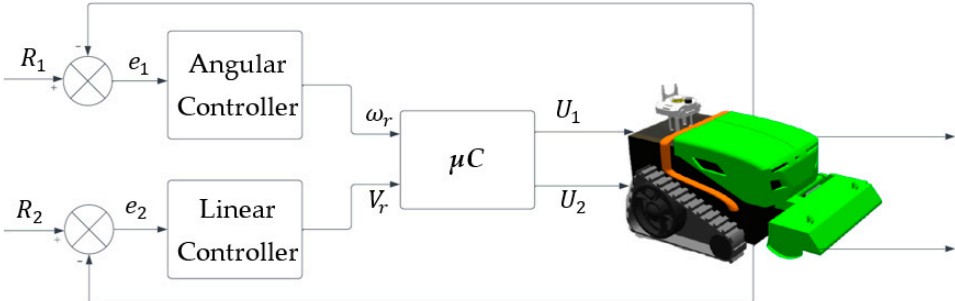

**Figure 9.** Robot Controller Architecture.

To proceed with the control of the tractor, tests were conducted to verify whether the commands sent by the ROS system resulted in the desired movements of the LV600 PRO. Figure 10 illustrates the operating principle for different voltage values VL and VR (left and right track, respectively) in the robot in question.

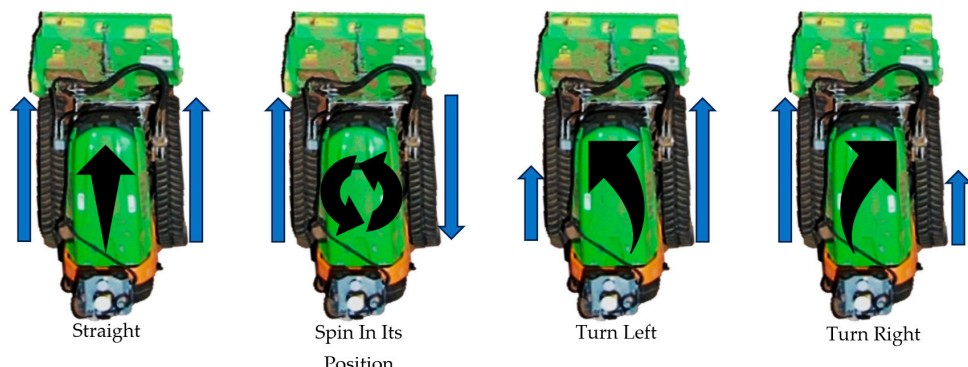

**Figure 10.** Possible movement of the tractor's tracks.

The controllers are based on classical PID control with variations in error variables. In one approach, the robot's orientation is calculated concerning the goal, involving two loops, one for distance error and another for orientation error. The distance loop determines the initial reference orientation by multiplying the distance error with a proportional component. The orientation loop controls the robot's angular movement based on this reference. In an alternative method, attractive vector fields are generated using destination coordinates. The control of the tractor's angular component relies on the vector sum of these fields.

*2.6. Sensors*

The forest environment is unstructured and requires the use of a diverse set of sensors to enable the robot's navigation. Obtaining 3D-perception information is crucial and can be achieved through various sensors such as stereo cameras, 3D LiDAR, and sensor fusion of different devices, including GPS and IMU, among others [32]. Sensor information not only enables the movement of the robotic system but also allows for assessing terrain conditions and performing precise cutting of the vegetation to be removed. This challenge is even more pronounced in rugged terrains with dense vegetation [8].

For the development of a flexible, reliable, and easily maintainable autonomous mobile robot, it is crucial to establish a robust, simple, and modular architecture encompassing both sensors and actuators, along with computers responsible for executing algorithms to perform tasks [19]. However, an additional complexity arises, especially when some position sensors, such as GPS/RTK, do not provide accurate enough data, making it challenging to obtain the correct position of the system, especially in areas where cleaning tasks are performed [25]. Another challenge arises in the nature of the soil, often filled with loose elements such as rocks and tree trunks, as well as cavities and holes. These adverse

conditions can result in slippage or even the locking of the robotic system. Additionally, unlike agricultural robots, which are sheltered during storms, forest robots operate after adverse weather events. This implies the need for communication and locomotion systems adapted to these challenging environments, as well as the ability to remain operational even after storms or fires [8].

### 2.6.1. Filter Data

In forest environments, the presence of noise in sensor-acquired data can create significant obstacles to the effective operation of Autonomous Mobile Robots (AMRs). This interference can originate from various sources, including vegetation density, fluctuations in lighting conditions, sudden changes in terrain, and other unforeseen factors. Consequently, the implementation of filters becomes crucial to address these uncertainties and enhance the reliability of the obtained sensorial information. By applying appropriate filters, AMRs can identify more precise patterns in sensorial information, adapting more effectively to challenging environmental conditions. These filters not only smooth out undesired variations in sensor signals but also contribute to the stability and consistency of AMR operations. In forest environments, where topography and vegetation can change rapidly, the ability to filter and accurately interpret sensor data is crucial to ensuring the safe and efficient navigation of these robots [33].

Thus, the careful integration of filters into AMRs' sensor systems not only helps overcome sensor challenges but also enables the execution of specific tasks, such as vegetation cutting, in a more autonomous and precise manner. This sensory adaptability is fundamental for the operational effectiveness of AMRs in dynamic and unpredictable forest environments.

To improve the dependability of sensor data, it is crucial to tackle noise. The Exponentially Weighted Moving Average Filter (EMAF) was chosen for its capacity to give precedence to recent information, an important aspect for swift responses to fluctuations in the input data [34].

### 2.6.2. Camera Detection

To visually detect and classify obstacles, two RGBD D435i cameras from Realsense and two thermal cameras from FLIR ADK were used. By integrating the You Only Look Once (YOLO) algorithm, obstacle detection and identification became fast and direct. This algorithm operates with the entire image as input to a neural network, eliminating the need for multiple complex steps.

YOLO directly determines the location of bounding boxes for objects in the image, classifying each detected object and providing accurate information about the position and class of each identified element, as illustrated in Figure 11 [35]. Image information is used to project each bounding box, containing five values: coordinates, width, height of the box, and confidence associated with each detection. These values are related to a grid of cells covering the image, with each cell, located at the center of a bounding box, containing information on how to detect a specific object [36].

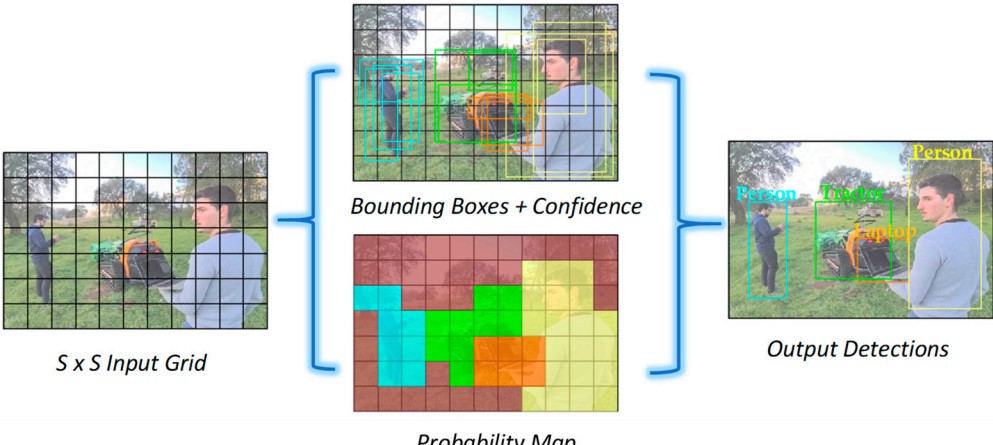

**Figure 11.** YOLO algorithm representation.

YOLO utilizes information from the entire image to detect objects at once, accelerating the process compared to more complex approaches [35]. It is crucial to highlight that the performance of YOLO depends on the training dataset and its quality. To increase data robustness, both available camera technologies were chosen. Thermal sensors can capture images by detecting the thermal energy emitted by objects. They are highly advantageous because they are not affected by changes in lighting conditions and are suitable for various light levels and weather conditions [37]. In the domain of security applications, thermal cameras are of immense importance, especially in challenging weather situations such as rain and fog, where traditional RGB cameras may encounter difficulties. Moreover, they are valuable in situations of total darkness when conventional cameras cannot operate effectively. On the other hand, infrared cameras provide less detail compared to visible light cameras because colors captured in the visible spectrum offer richer and easily interpretable information [38].

2.6.3. LiDAR Detection

LIDAR sensors play a crucial role in autonomous navigation, providing not only an accurate 3D view of the surrounding environment but also generating a point cloud that enables data manipulation for object detection. This allows the identification and measurement of these points in real time, thus creating a precise map of a constantly changing environment and enabling safe navigation. Due to their precision in distance measurement, LIDAR sensors can detect and avoid objects up to 200 m away, even in challenging weather and lighting conditions [39]. In Figure 12, we observe an illustration of how the lasers from the LIDAR propagate, with this LIDAR having 16 lasers.

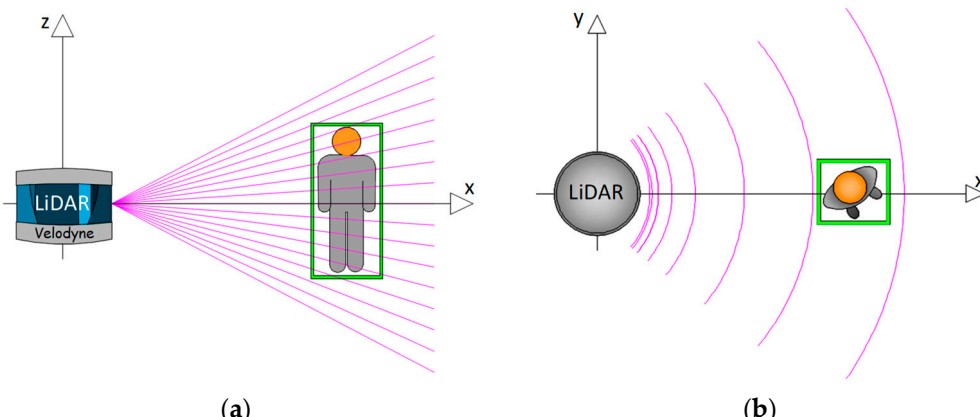

**Figure 12.** Illustration of LiDAR detection. (**a**) Axis ZX. (**b**) Axis YX.

## 3. Results

The practical tests aim to validate all the implemented algorithms and functionalities in the sensors. Initially, machine tests were conducted through the teleoperation system, allowing an operator to control the machine. In these tests, the orientation and position of the robot were evaluated, also validating the detection algorithm through the vision system, with a focus on the adaptive clustering algorithm associated with LIDAR. In subsequent phases, navigation experiments were conducted to validate the autonomous behavior of the machine. All these experiences were carried out in a real environment, as illustrated in Figure 13, providing real conditions to test and enhance the system's performance.

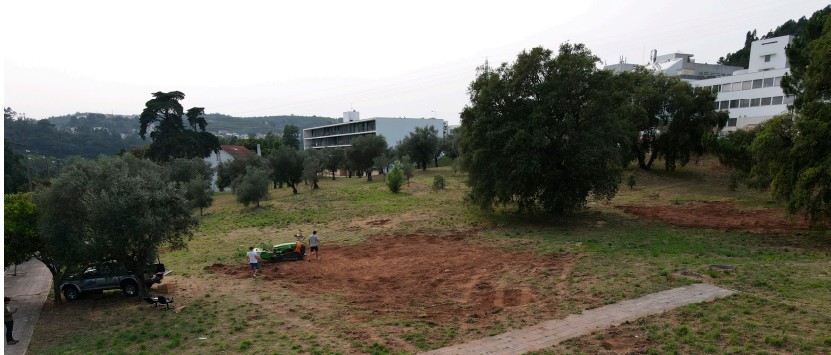

**Figure 13.** Experimental setup for algorithms validation.

### 3.1. Controller and Sensors Integration

For the sensor system to be able to control the machine in question, the process began by testing all the inputs of the machine's interface to understand the voltage and current values needed for each machine action. This means that depending on the voltage applied to the terminals of the tractor's plug, it controls solenoid valves already present in the original tractor. Figure 14 shows the interface of the machine where these variables were measured.

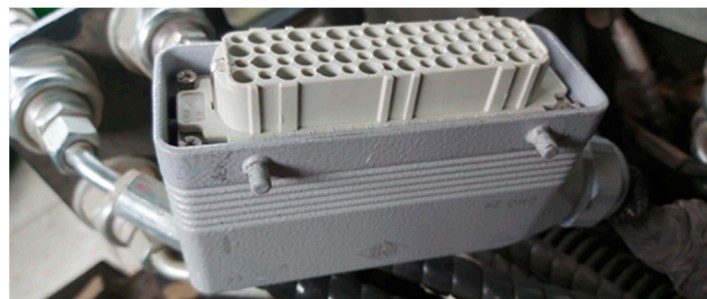

**Figure 14.** Universal forest machine plug.

The tractor comes with default corresponding pins, but it was found that not all the pins' present are necessary for controlling the tractor. In Figure 15, we can observe THE initial phase of the tractor's pin layout as well as the final phase of this layout after selecting the required inputs and outputs for its proper control.

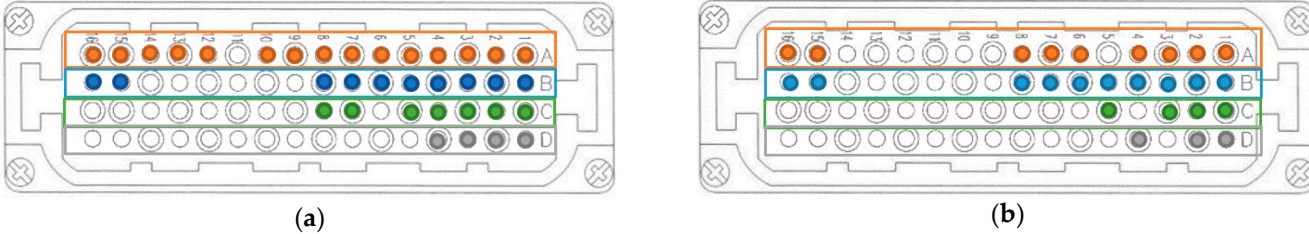

<div align="center">(<b>a</b>)         (<b>b</b>)</div>

**Figure 15.** Plug Terminals (**a**) Before testing. (**b**) After testing and remove the unnecessary pins.

After the accurate measurement of each of these pins, it was possible to develop a code that enabled autonomous control of the machine. To achieve this, the sensor system was developed, as shown in Figure 16.

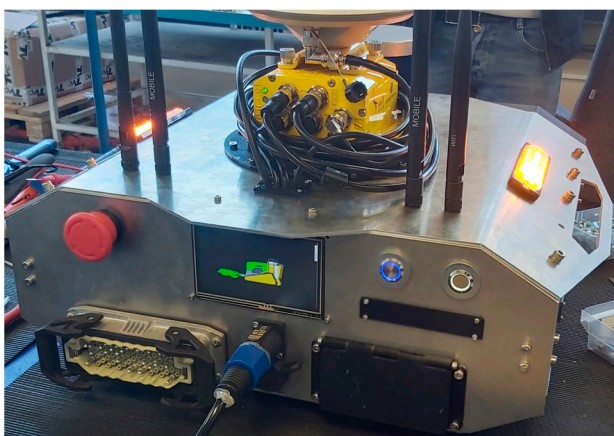

**Figure 16.** Activation of the sensor system.

Concluding this sensor module, testing of the robot was initiated. The connection between the module and the tractor was established using a 41-conductor cable. In Figure 17, we can observe the connection between the two in the yellow boxes.

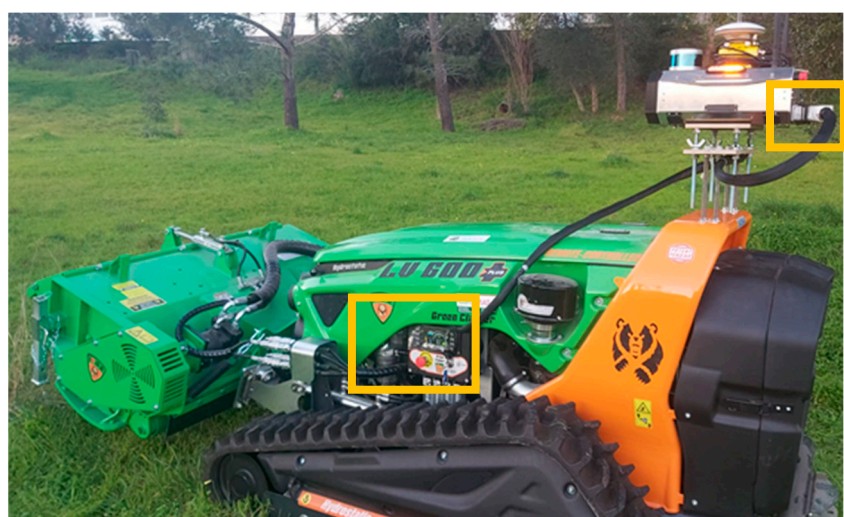

**Figure 17.** Connection of the sensor system into the forestry machine.

*3.2. Camera Detection*

In Figure 18, it is possible to observe a forest scenario where thermal and RGBD cameras detect the presence of a person amidst the vegetation using the YOLO algorithm. The use of this algorithm proves to be crucial in forest environments, where the quick and

accurate identification of objects, such as people or obstacles, is of utmost importance. The implementation of this algorithm significantly contributes to the effectiveness and safety of the robotic system's operations in this specific context. The robot is configured to detect a person at a distance of less than 7 m, automatically coming to a halt. It remains stationary until the distance exceeds 7 m to resume its operation.

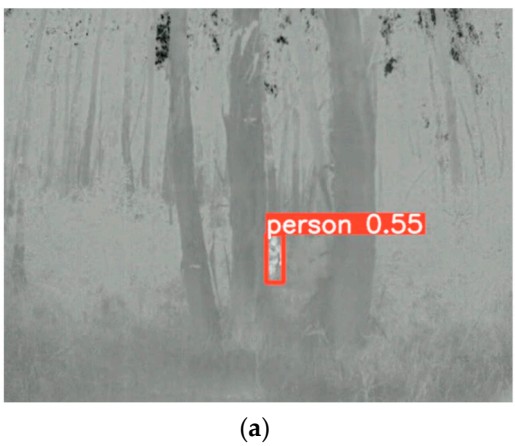
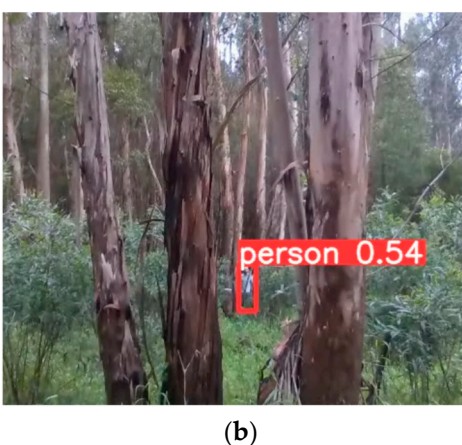

(**a**)                                                                                           (**b**)

**Figure 18.** Comparison test between cameras. (**a**) Thermal Camera. (**b**) RGBD Camera.

A neural network training process was conducted to enable the detection of trees with the algorithm, as these represent obstacles that the robot should not cut. Specific datasets designed for tree identification were used, comprising thousands of images with annotated trunks [40]. After completing the neural network training, the algorithm underwent additional tests, and the results obtained are illustrated in Figure 19.

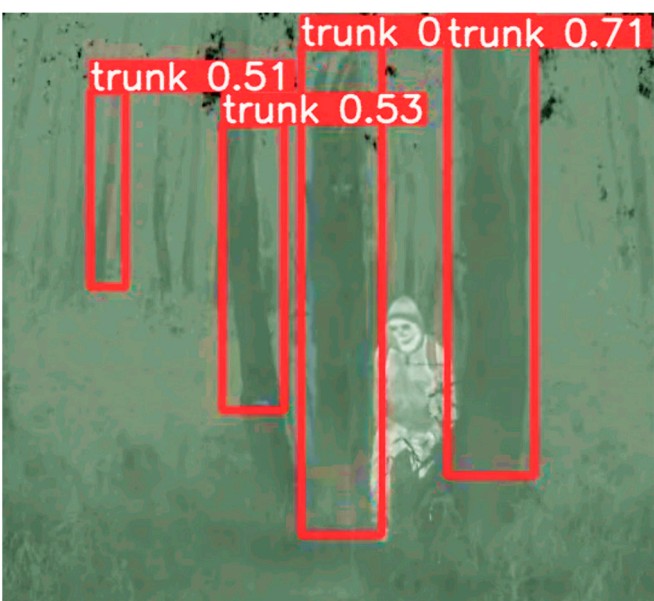

**Figure 19.** Tree Detection after a new neural training.

### 3.3. LiDAR Detection

The PointCloud, in conjunction with the clustering algorithm, plays a crucial role in the robot's perception of its environment. By collecting data through LiDAR, the algorithm identifies clusters of points representing objects such as people and trees. These clusters are then mapped into bounding boxes, facilitating the interpretation and manipulation of the data. This information is published, allowing for other modules and systems of the robot, such as path planning, to use these data to make real-time decisions. Providing the

distance to each obstacle is crucial for the autonomous and safe navigation of the robot in challenging environments such as forests. LiDAR data and obstacle identification through clustering are key elements to ensure that the robot can dynamically perceive and react to its surroundings, avoiding collisions and optimizing its navigation route. In Figure 20, we can observe the implementation of these algorithms in the sensor module, where the obstacles present in the robot's operating environment are referenced as green boxes.

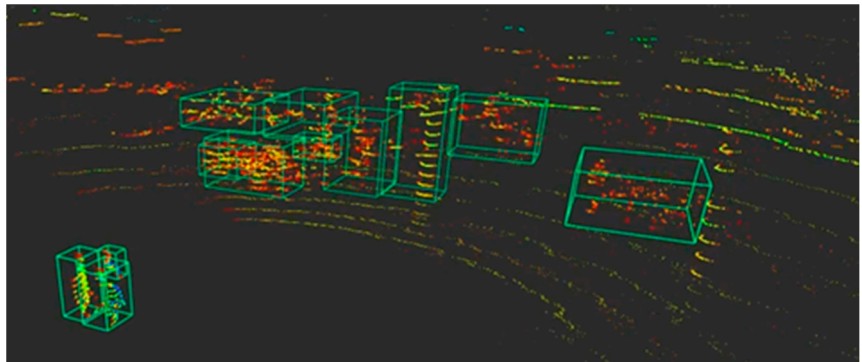

**Figure 20.** Object Clustering in a real-world scenario.

*3.4. Sensors Filter*

To assess the effectiveness of the filters, two sets of tests were conducted for each. The first set occurred under stationary conditions, while the second set during a state transition. The filters implemented are referred to as Low Pass Filter (LPF) and Exponential Moving Average Filter (EMAF). In the first test, parameters such as Maximum Absolute Error, Mean Absolute Error, and Variance were calculated. The objective was to keep the magnetometer (a sensor measuring Earth's magnetic fields) at rest, without vibration or rotation, recording values over 0.3 s. In the second test, data acquisition took place during a state transition. For this, the magnetometer was manually rotated on a flat surface, covering approximately 120°, to observe the response to an impulse. Each rotation took approximately 1 s to complete. The purpose of these tests was to determine the most suitable order and cutoff frequency for the filter, aiming for a quick response, and identify the buffer size that best adapts to state changes. All tests were conducted in a laboratory environment, with the Duro Inertial positioned on a flat surface. In Figure 21, differences associated with the presence of a filter in receiving data from the magnetometer can be observed.

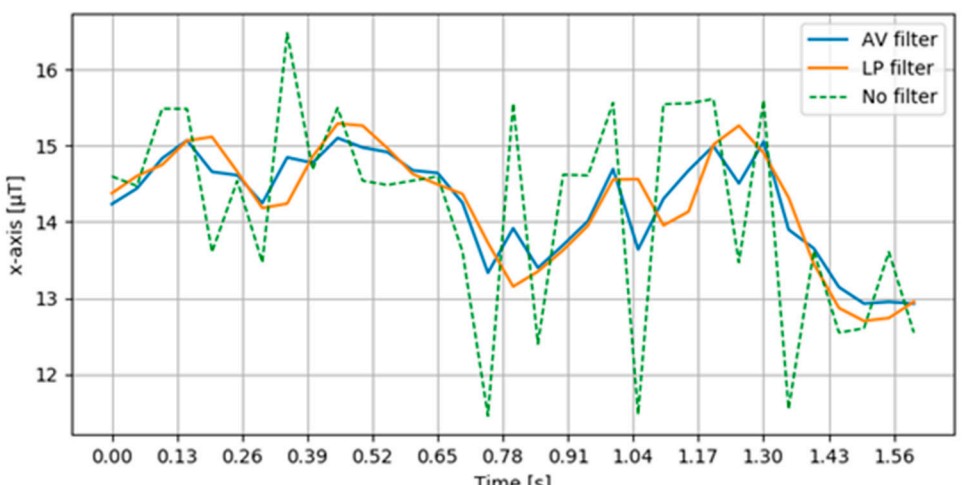

**Figure 21.** Comparison of data with and without filtering between AV and LP filter.

Not only for the magnetometer but also for the data received from the GPS, a filtering process was applied. The graph in Figure 22 provides a clear view of the superior per-

formance of the Kalman Filter in filtering GPS data. Noticeably, there is a reduction in the maximum distance value concerning the reference trajectory when the filter is used compared to its absence. Additionally, the filter shows a lower RMS value, indicating higher reliability in the results. The SMA filter also exhibits a reduced RMS value, although not superior to that presented by the Kalman filter.

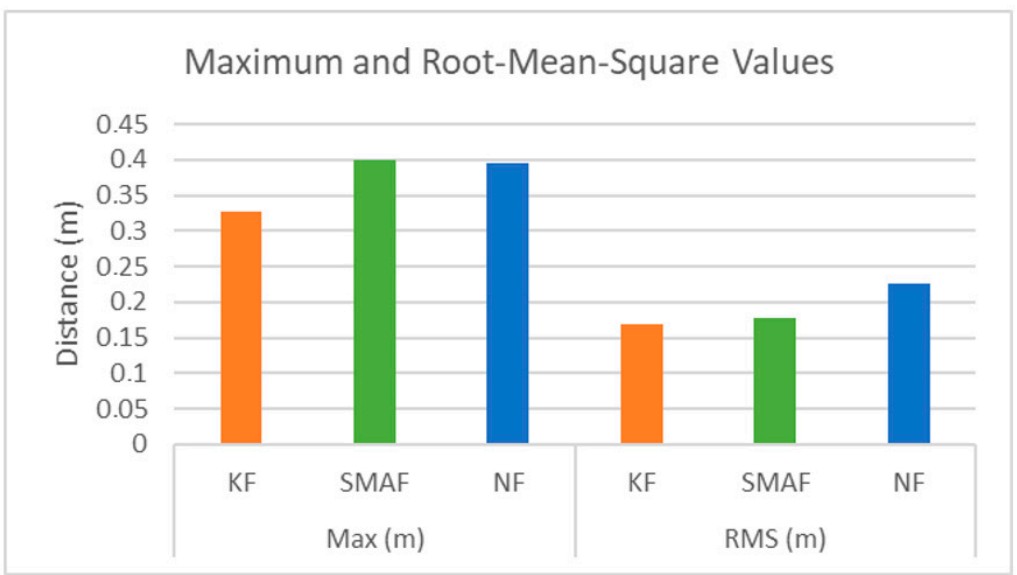

**Figure 22.** Comparison of data with and without filtering between KF and SMAF.

After the filtration of sensor data, the orientation of the sensor system was calculated. In Figure 23, we can observe that the robot, oriented approximately to the West, is following what is observed on the computer screen.

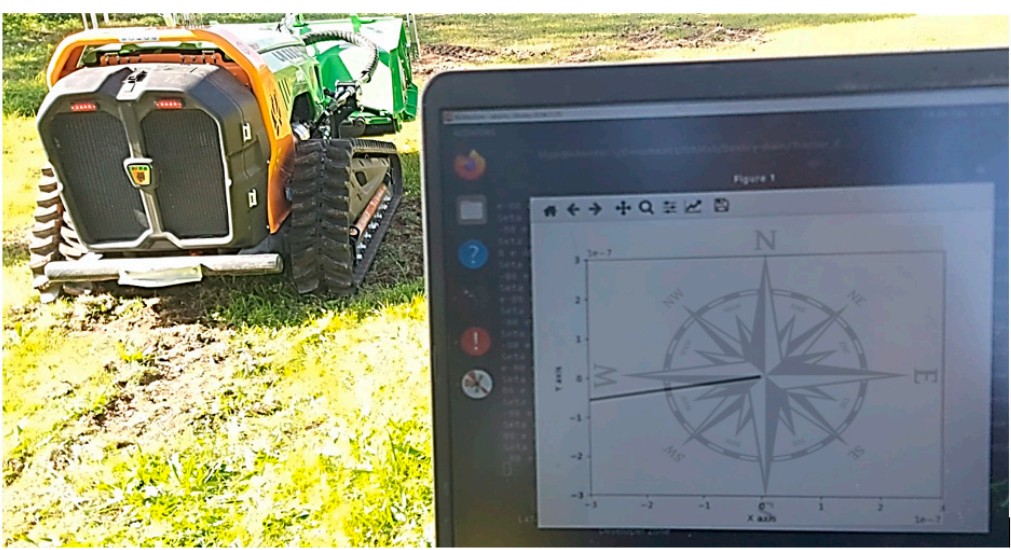

**Figure 23.** Experimental test for machine orientation.

*3.5. Navigation*

To confirm the ability to navigate and control the machine, practical tests were carried out with different navigation methods and a comparison between these methods to determine the most effective method was performed. A real scenario was simulated where the robot traveled through a series of points: (0,0), (10,0), (10,2), (0,2), (0,4), (10,4), (10,6), (0,6), and (0,0).

Figure 24 illustrates the data obtained along this path.

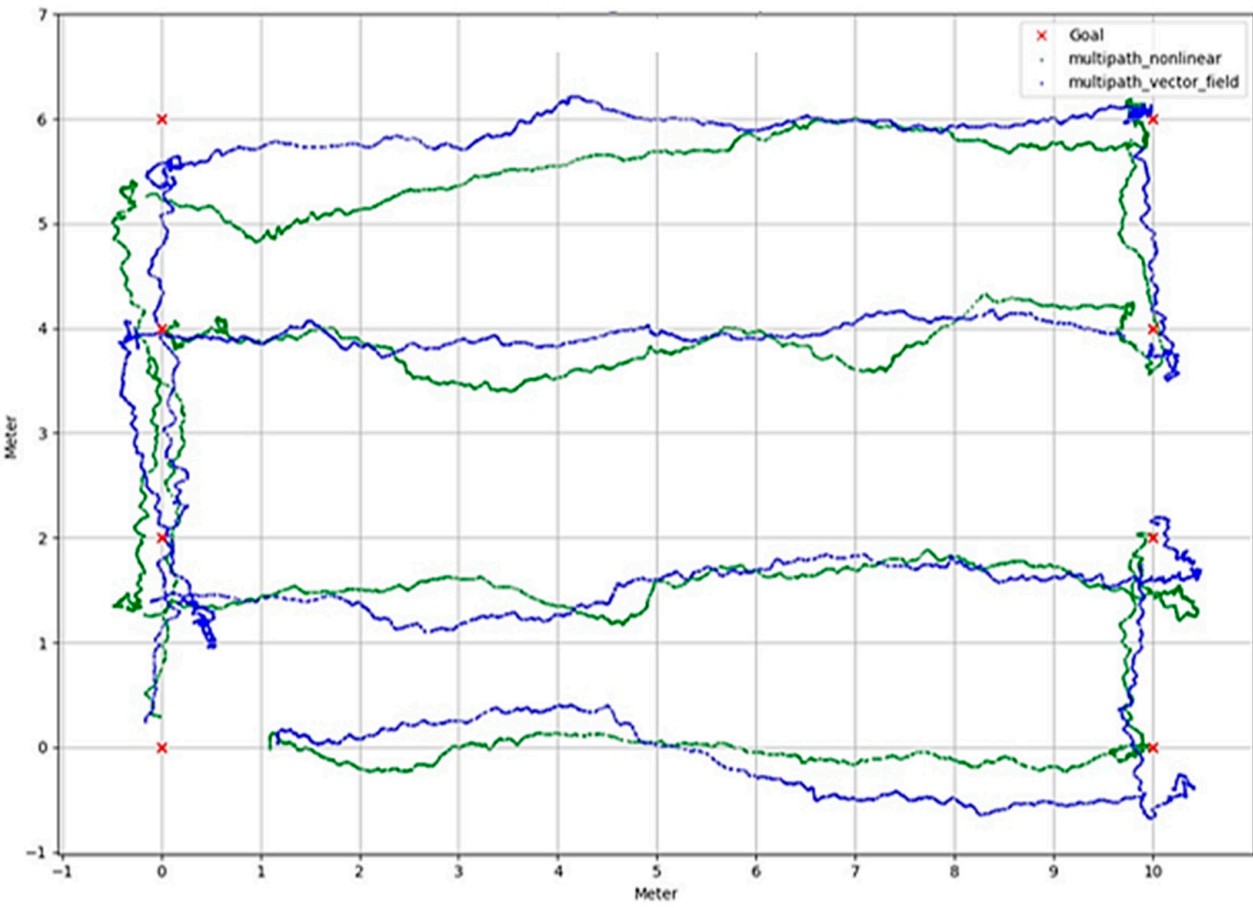

**Figure 24.** Combining the trajectories of both Controllers (1 and 2).

As depicted, all the points were successfully reached. For better visualization, these paths were separated and presented individually in Figure 25.

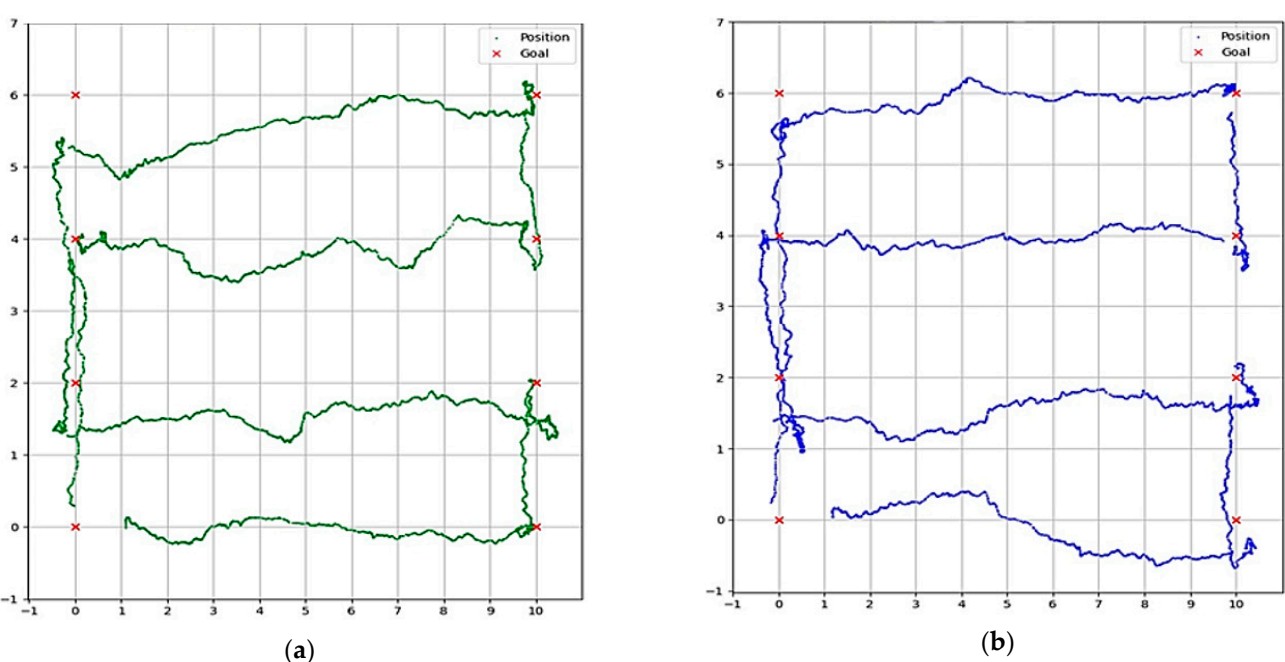

(**a**)                                                                                  (**b**)

**Figure 25.** Trajectories of each controller. (**a**) Nonlinear. (**b**) Vector Field.

Now that we have the data from these trajectories, it is possible to calculate the desired errors to draw a conclusion about which algorithm is better, as shown in Table 3.

**Table 3.** Errors and Navigation Time for Controllers 1 and 2.

| Algorithm | Mean [m] | Standard Deviation [m] | Maximum [m] | Navigation Time [s] |
|:---:|:---:|:---:|:---:|:---:|
| 1 | 0.28 | 0.25 | 1.16 | 175.03 |
| 2 | 0.25 | 0.21 | 0.93 | 185.5 |

With these results, despite Controller 2 having a navigation time of approximately 10 s longer than Controller 1, all the errors involved in navigation are smaller. Thus, it is concluded that Controller 2 is the controller that generates the least error in this navigation.

### 3.6. Obstacle Avoidance

Two obstacle avoidance methods were employed: the A* algorithm, a pathfinding algorithm that continuously explores unexplored locations in a graph, and the Artificial Potential Field (APF), which simulates the robot's environment by using attractive and repulsive forces. The A* algorithm halts when the target is reached [41], while the APF method involves repulsive forces from obstacles and an attractive force from the target [42]. Successful simulation tests were conducted for both algorithms, as depicted in Figure 26.

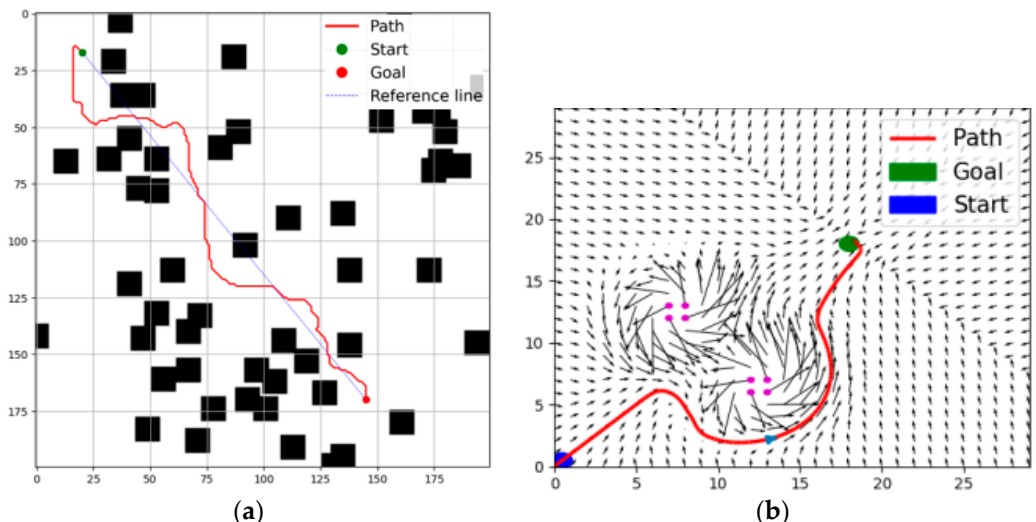

(**a**)           (**b**)

**Figure 26.** Obstacle avoidance algorithms (**a**) A* (**b**) Vector Field.

After validating the algorithms, a practical experimental test was conducted, where, as shown in Figure 27, we can observe the paths taken by the tractor using the following algorithms.

To compare these algorithms, error tests were conducted to determine which performed better. The "X" represents the obstacle around which the tractor navigated. Table 4 presents the results of the values obtained from these algorithmic comparisons.

**Table 4.** Errors and Navigation Time for Algorithms A* and VF.

| Algorithm | Mean [m] | Standard Deviation [m] | Maximum [m] | Navigation Time [s] |
|:---:|:---:|:---:|:---:|:---:|
| A* | 1.70 | 1.59 | 4.96 | 32.67 |
| VF | 1.54 | 1.53 | 5.62 | 112.47 |

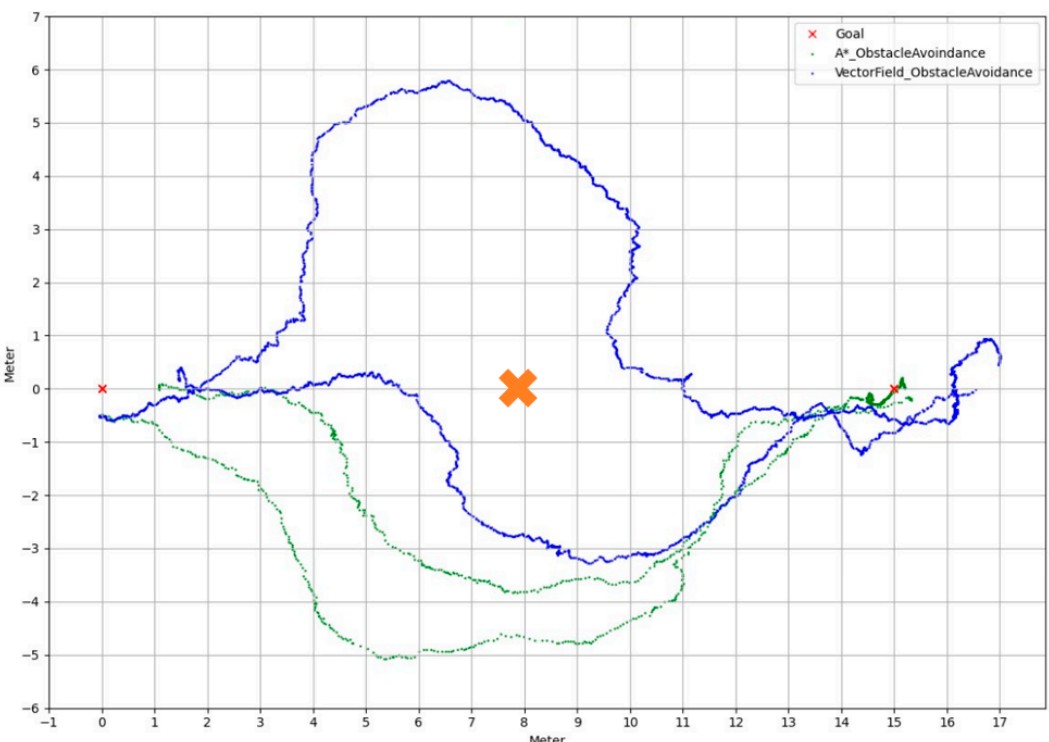

**Figure 27.** Obstacle avoidance paths of both algorithms.

It is noted that the A* algorithm outperforms the VF algorithm. Its navigation time is almost four times faster than the VF algorithm, with a lower maximum error, while the remaining aspects are relatively comparable. In conclusion, the A* algorithm emerges as the winner in this study.

## 4. Discussion

The development of this sensorial component represents a significant milestone in the quest for efficient navigation of autonomous machines in challenging environments such as forests [43]. The quality of data from sensors plays a crucial role in this process, underscoring the importance of effective filters to deal with the complexities of the environment [34].

The use of different control and object detection methods underscores the multifaceted approach adopted in the project. The diversity of techniques employed highlights the complexity of the task at hand and emphasizes the importance of an integrated approach to ensure optimal results. The discussion on conducting tests involving RGBD and thermal cameras emphasizes the focus on safety and efficient navigation. The combination of these technologies offers a more comprehensive view of the environment, overcoming the limitations of each individual sensor [44]. The mention of the ability of RGBD cameras to calculate distances and their complementarity with LiDAR distances highlights the synergy between different sensory modalities. This not only reinforces the robustness of the system but also underscores the importance of data fusion to obtain more precise and comprehensive information about the surrounding environment. The discussion on filters applied to sensors highlights the need to deal with the inherent noise in sensor data. The choice of filters, such as LPF and EMAF for the magnetometer, and the use of the Kalman filter for GPS, highlights the effectiveness of these techniques in improving the accuracy and reliability of data [34].

Finally, the use of two controllers in navigation demonstrates a holistic approach to machine control. The prevalence of vector control suggests a better suitability for this specific environment, emphasizing the importance of choosing the most appropriate control method for the conditions of the task. In summary, the article comprehensively

addresses the importance of the sensorial component, presenting results and decisions that significantly contribute to the effectiveness of the machine's navigation system in challenging forest environments.

### 5. Conclusions

In summary, although the forestry machine is still in the development phase, the collected data already indicates promising results. The incorporation of various navigation methods and detection approaches demonstrates a continuous and innovative commitment. However, the need for improvements is recognized, especially at the level of the implemented code and a more in-depth analysis of the peculiarities of the forest environment. The ongoing evolution of the machine will require constant improvements in the code to optimize operational efficiency. Additionally, it is crucial to conduct a more detailed investigation into the complexities of the forest environment to ensure that the machine performs its functions even more effectively in the future. Therefore, the path toward the successful implementation of this machine embraces a holistic approach, integrating technological advancements with a deep understanding of the subtleties of the operating environment. By maintaining a constant commitment to technical and scientific improvement, it is possible to anticipate more efficient and sustainable solutions to meet the demands of the forestry sector. The development of SLAM using vision-based methods would be highly interesting, especially since the robot lacks wheel encoders, making odometry challenging. Visual odometry could provide a viable solution. Regarding control, starting with more modern methods such as fuzzy logic and predictive control could be beneficial. Testing obstacle avoidance in more complex environments would provide valuable insights for the case study. Additionally, conducting experimental tests with intentionally induced sensor failures to assess and test sensor redundancy would be valuable, forcing the system to navigate even in the presence of faults.

**Author Contributions:** Methodology, T.G., T.P. and N.F.F.; Software, T.G., T.P. and F.D.G.; Validation, N.F.F.; Formal analysis, N.F.F.; Investigation, T.G., T.P. and F.D.G.; Resources, C.V. and F.D.G.; Data curation, T.G. and N.F.F.; Writing—original draft, T.G.; Writing—review & editing, C.V. and N.F.F.; Visualization, T.P. and N.F.F.; Supervision, C.V.; Project administration, C.V.; Funding acquisition, C.V. All authors have read and agreed to the published version of the manuscript.

**Funding:** The present work was funded through the projects E-Forest—Multi-agent Autonomous Electric Robotic Forest Management Framework, ref. POCI-01-0247-FEDER-047104, and F4F— Forest for Future, ref. CENTRO-08-5864-FSE-000031, co-financed by European Funds through the programs Compete 2020 and Portugal 2020, as well as by FCT (Fundação para a Ciência e a Tecnologia) under https://doi.org/10.54499/UIDP/00760/2020, project LA/P/0079/2020, DOI: 10.54499/LA/P/0079/2020, and https://doi.org/10.54499/UIDB/00760/2020 (accessed on 19 October 2023). Additionally, financial support was received from the Polytechnic Institute of Coimbra under the scope of Regulamento de Apoio à Publicação Científica dos Estudantes do Instituto Politécnico de Coimbra (Despacho n.º 5545/2020).

**Data Availability Statement:** The following supporting information can be downloaded at: https://susy.mdpi.com/user/submission/video/8e3dc6b9faf216ea2ecf327bd1afd602.

**Conflicts of Interest:** The authors declare no conflict of interest.

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
