# Peer review of "Robots for Forest Maintenance"

_forests, doi:10.3390/f15020381_

Round 1

Reviewer 1 Report

Comments and Suggestions for Authors

The Introduction needs additional reinforcement from scholarship. For example, paragraphs two and three in the Introduction were not supported by contemporary literature. 

Likewise the Discussion and Conclusions sections were not supported, citations not included, to the author's provided citations in the earlier portions of the manuscript or any additional literature. The lack of the connectivity between the Discussions and Conclusions to contemporary literature is the fundamental flaw of the manuscript. Twenty-four total citations in the References section is indicative of the light literature review depth demonstrated in this manuscript. 

Finally, the authors should indicated the role of industry, academic institutions, and government in the development and/or dissemination of the technology to stakeholders or actors (i.e. landowners, companies, decision makers, etc.) in forestry systems. This would greatly enhance the scholarship's impact on science and practice, improve it's translational research, and indicate the importance of the science beyond the study region. 

Author Response

Dear Reviewer,

I hope this message finds you well. I am submitting the revised article with the changes I believe address your feedback. Regarding the component you mentioned, my responses to your alterations are indicated by blue underlining.

Best regards,
Tiago Gameiro

Reviewer 2 Report

Comments and Suggestions for Authors

The title of the article and research objectives have been written very well. Therefore, readers hope to get a complete exposure to robot design, both in terms of hardware, software, and methods and algorithms. Furthermore, testing and implementation will provide an overview of the robot's performance, so that there is an overview of its implementation.

It's a shame that the design is still global and partial. There is no narrative or explanation about how the method or algorithm is implemented in the program. Robot design involves many sensors with different characteristics. Of course it would be very interesting if the author revealed a method for dealing with it. Especially in handling data from sensors, both formats, standards and protocols used. Given the complexity of the sensors installed and the actions the robot performs (as shown in figure 3 page 5), the question arises about the type of processor used. If (mentioned in line 104) the robot is controlled using an On Board CPU (perhaps what is meant is a Single Board Computer), then many questions arise regarding the specifications. What is the word length? What is the memory capacity? What is the clock speed? What bus is used for communication with sensors and actuators? Moreover, in lines number 154 to 158 it is stated that the navigation of this robot is using online maps.

The most obvious weakness is the lack of integration of all sensors and actuatures with the processor, as well as integration tests. Next, the robot, which is equipped with a control system, sensors and actuators, is tested in various scenarios designed to measure the robot's performance.

This article has been written with good English expressions. The choice of terms in this writing really deserves appreciation. However, there are still many sentences written very long, making it difficult for readers to understand this article.

Another weakness, image 20 on page 14 is still written in a language that is not English.

Comments on the Quality of English Language

This article has been written with good English expressions. The choice of terms in this writing really deserves appreciation. However, there are still many sentences written very long, making it difficult for readers to understand this article.

Author Response

Dear Reviewer,

I hope this message finds you well. I am submitting the revised article with the changes I believe address your feedback. Regarding the component you mentioned, my responses to your alterations are indicated by green underlining.

Best regards,
Tiago Gameiro

Reviewer 3 Report

Comments and Suggestions for Authors

The topic is interesting but there are many topics that need to be addressed.

1-The machine that you picked is remotely controlled and powered by engine, how have you automated it? No detail is provided on that topic

2- No detail is provided about the sensor fusion.

3- The novel technical contributions should be explicitly stated.

4- Path planner is mentioned, but no detail is provided about path planning algorithm.

5- The state machine in Figure 5 needs further clarification. What do you mean by arrived? Is it reached? why two meters front and one meter back? If there are any obstacles in front, how would you sense them?

6- No detail is provided about the kinematics of the vehicle.

7- what are R1 and R2 in Figure 8? they are already known? what are the feedbacks provided by the vehicle. which controller is being used? I am not talking about micro-controller.

8- The role of each sensor being used must be explicitly specified.

9- What would be the response of the forest machine once the person is detected?

10- The figure captions and titles must all be in English language.

11- The results provided in Figure 22 and 23 are very hard to understand. They should be redrawn with clear understanding.

Comments on the Quality of English Language

moderate changes are needed.

Author Response

Subject: Submission of Revised Article

Dear Reviewer,

I hope this message finds you well. I am submitting the revised article with the changes I believe address your feedback. Regarding the component you mentioned, my responses to your alterations are indicated by yellow underlining.

Best regards,
Tiago Gameiro

Reviewer 4 Report

Comments and Suggestions for Authors

This manuscript presents a novel technology for developing robotic systems tailored for forest management. The research is compelling and yields valuable insights; however, the document contains several deficiencies that need addressing to ensure the outcome aligns with the esteemed caliber of the publication.

(1) The experimental section would benefit from the inclusion of more quantitative metrics to assess system performance, such as positioning accuracy and the success rate of obstacle avoidance.

(2) A more comprehensive exposition of the algorithms and control systems, including their working principles, benefits, and constraints, is necessary. An in-depth discourse on the design of response controllers within the framework of multiple input multiple output (MIMO) control systems could enrich the paper.

(3) The manuscript underscores the significance of sensor data filtration. It should also elucidate the rationale behind the selection of specific filters and their consequent effect on data reliability and precision.

(4) The algorithmic description requires elaboration. Incorporating algorithmic flowcharts would greatly enhance the comprehensibility of the methodologies for readers.

(5) The conclusion should extend its discussion to potential future enhancements and developments. It would be constructive to ponder on avenues such as testing in more intricate scenarios, refining algorithms, or integrating additional sensors.

(6) The authors may add more state-of-art computer vision application articles for the integrity of the manuscript (Optimization strategies of fruit detection to overcome the challenge of unstructured background in field orchard environment: a review; Precision Agriculture. A Performance Analysis of a Litchi Picking Robot System for Actively Removing Obstructions, Using an Artificial Intelligence Algorithm; Agronomy. Transforming unmanned pineapple picking with spatio-temporal convolutional neural networks; Computers and Electronics in Agriculture.).

Author Response

Dear Reviewer,

I hope this message finds you well. I am submitting the revised article with the changes I believe address your feedback. Regarding the component you mentioned, my responses to your alterations are indicated by gray underlining.

Best regards,
Tiago Gameiro

Round 2

Reviewer 1 Report

Comments and Suggestions for Authors

Authors made the necessary revisions and additions I provided in the first review. The scholarship is more indicative of the science's impact on practice and how said practices can be improved due the scholarship presented. The authors additionally provided delineations to the extent their scholarship can be replicated and potentially impactful beyond the researched regions specifically in reducing and addressing forest fires. 

Author Response

Thank you for comments

Reviewer 2 Report

Comments and Suggestions for Authors

The revised article just great. 

Author Response

Thank you for comments

Reviewer 3 Report

Comments and Suggestions for Authors

Previous comments are still valid.

Comments on the Quality of English Language

moderate language changes are required.

Author Response

The responses are highlighted in yellow, please see the attached file.
